# A multi-year science research or engineering experience in high school gives women confidence to continue in the STEM pipeline or seek advancement in other fields: A 20-year longitudinal study

**Patricia K. Hunt, Michelle Dong** , **Crystal M. Miller** *

Hathaway Brown School, Shaker Heights, Ohio, United States of America

* cmiller@hb.edu

## Abstract

There remains a large gender imbalance in the science, technology, engineering and mathematics (STEM) workforce deriving from a leaky pipeline where women start losing interest and confidence in science and engineering as early as primary school. To address this disparity, the Science Research & Engineering Program (SREP) at Hathaway Brown School was established in 1998 to engage and expose their all-female high school students to STEM fields through an internship-like multi-year research experience at partnering institutions. We compare data from existing Hathaway Brown School SREP alumnae records from 1998–2018 (*n = 495*) to Non-SREP students and national datasets (National Center for Educational Statistics, National Science Foundation, and US Census data) to assess how SREP participation may influence persistence in the STEM pipeline and whether SREP alumnae attribute differences in these outcomes to the confidence and skill sets they learned from the SREP experience. The results reveal that women who participate in the SREP are more likely to pursue a major in a STEM field and continue on to a STEM occupation compared to non-SREP students, national female averages, and national subsets. Participants attribute their outcomes to an increase in confidence, establishment of technical and professional skills, and other traits strengthened through the SREP experience. These data suggest that implementing similar experiential programs for women in science and engineering at the high school stage could be a promising way to combat the remaining gender gap in STEM fields.

## Introduction

In the last 20 years, substantial gains have been made in the effort to increase the number of women in STEM (science, technology, engineering, and mathematics) fields [1, 2]. While both boys and girls in middle and high school take roughly equal numbers of math and science courses and earn similar grades [3], they diverge in selecting college majors, deviate further in completion of those selected majors, and women's numbers decline more at the graduate level

**Data Availability Statement:** Data cannot be shared publicly because the small sample size from each graduation year could allow recognition of

individual identity. Data are available from Hathaway Brown School (contact srepwebmaster@hb.edu) for researchers who meet the criteria for access to confidential data.

**Funding:** The author(s) received no specific funding for this work.

**Competing interests:** I have read the journal's policy and the authors of this manuscript have the following competing interests: Crystal M Miller is an employee of Hathaway Brown School.

and in the workplace in many STEM fields [4–7]. Even among the life sciences, one of the only STEM disciplines where women have achieved parity, men outnumber women in leadership and tenure-track positions [4, 8]. Nationally, the US educational system is not producing a sufficient STEM workforce, impacting economic productivity and competitive innovation [9]. The most obvious solution to fill this talent gap would be to focus on diversifying the pool to include more women and other minority groups in STEM fields, but a solution on how to accomplish this task is still forthcoming [10].

Starting in 1998, a multi-year STEM research experience has been ongoing at Hathaway Brown School (HB), an all-female private school in Shaker Heights, OH. The program was established to encourage the pursuit of STEM majors for women and increase confidence and empowerment by giving this all-female population of high school students the opportunity of an internship-like research experience in a STEM field by partnering students with researchers at local universities and institutions. Based on outcomes from the Science Research & Engineering Program (SREP) alumnae from 1999–2018, we demonstrate that the SREP, and similar experiential programs for high school girls, may be one step to increasing women in STEM fields and bridging the confidence gap that drives women away from STEM and leadership roles.

## Pipeline leakiness

The STEM pipeline is a metaphor for the pathway students enter in early education until college graduation that intends for a finalized destination in a STEM occupation. From 1998–2016, women earned 50.2% of STEM bachelor's degrees [11, 12]. Despite some fluctuations, on average, women earned more degrees in the biological/agricultural sciences (55.6%) and psychology/social sciences (65.6%) compared to men, but are still vastly underrepresented in other STEM fields, most notably among engineering (19.6%), physical science (40.5%), and mathematics/computer science (33%) bachelor's degrees over this 18-year period [11, 12]. Among first-year students entering college in 2014, 37.5% of females declared a STEM major compared to 49% of males [13]. While the majority of students who major in STEM fields make that choice in high school [14], attrition rates in college to non-STEM majors have been linked to grades, specific major, and first relevant course [15]. Women are more likely than men to change from a STEM major to a Non-STEM major in response to poor performance even when quantitatively equal to their male counterparts, particularly in male-dominated STEM fields [16, 17]. It is essential to examine why these disparities exist in certain STEM fields and consider methods to increase the representation of women.

After college, women earn half or more of advanced degrees at every level overall; however, in STEM fields this varies largely by discipline. In 2017, women earned 43.6% and 45.2% of STEM Master's and doctoral degrees, respectively. Women are still separated by a wide margin from men for degrees in engineering, mathematics, computer sciences and the physical sciences (earning an average of 33.9% of Master's and 26.1% of doctoral in these three subjects combined) [18]. Finally, while the number of women with STEM degrees has doubled over the past two decades, the disparity of men outnumbering women in STEM employment has only narrowed modestly [18] demonstrating how leaks at various stages in the pipeline result in the shortage of women at the occupational level.

## The confidence gap

At all stages of the STEM pipeline, a consistent obstacle for women is confidence, whether it be their attitudes and expectations about mathematics and science careers as early as kindergarten [8, 19], the lack of confidence in high school girls tie to grades and performance, or imposter syndrome (the inability to believe one's success is deserved) in early or late career

[20, 21]. Women at all stages and in all fields, not just STEM, have lower confidence than men. Confidence has been defined as "the purity of action produced by a mind free of doubt" [22] and this confidence gap is a well-documented phenomenon [23].

As early as middle school and into high school, female students show less interest and confidence than male students in mathematics, even though they display similar levels of performance [24]. Male high school students rate themselves as more competent than females, even after adjustment for performance [25]. In college, the initial choice of a STEM major is significantly impacted by students' confidence in their academic and mathematical abilities, and this factor has been shown to directly affect a woman's choice of a STEM field [7]. Among engineering students, 'professional role confidence'—the perceived confidence in ability to successfully fulfill the responsibilities of a profession—contributes to gender gaps in attrition from engineering [26]. Even in the workplace women have lower self-confidence than men. In a 2011 study by the Institute of Leadership and Management, 50% of women managers reported feelings of self-doubt compared to less than a third of men [27]. The literature points to a strong link between a woman's perception about ability and her evaluation of her competency to complete a STEM major or go on to a STEM occupation, regardless of actual performance, another articulation of the confidence gap. Given that this confidence gap is especially prominent when it comes to math, physical sciences, and engineering, bridging that gap at a critical stage like high school could positively impact future pursuit and persistence in male-dominated STEM fields and reduce the leakage of women from the STEM pipeline. Numerous studies on where 'leakiness', or loss of women in the pathway, occurs have been ongoing and it has become clear that there is no one easy answer [28].

## High school is a key developmental stage in the STEM pipeline for women

In the context of women, the STEM pipeline has been under intense scrutiny in order to determine at what stages girls and, later women, gain their interest in STEM or leave the track to pursue a Non-STEM occupation [14, 29]. However, the end of middle school into high school seems to be a critical stage for STEM selection. Interest in STEM at the beginning of high school will set the precedent for whether it is pursued in later education [30]. Morgan et al., [2013] demonstrates undergraduate major choice is highly associated with occupational plans in high school, indicating that decisions on career have often been decided prior to college and the choice of major is just a formality along the way. Similar studies also indicate that an intent to major in a STEM field upon entering college is the strongest predictor of who will complete degrees in STEM [31, 32]. While interest in high school appears critical, it is one clear stage at which girls begin to decrease their interest in STEM fields [30, 33].

A large number of explanations for this decrease have been identified at the level of secondary school. Most prominent concerns include stereotype bias [34, 35], lack of confidence tied to grades and performance [20, 21], a difference in mentorship between the sexes, a shortage of female role models [36, 37], and social pressures and romantic interest [38, 39]. Interventions that have been shown to positively influence women toward a STEM major include taking AP courses [40], utilizing exploratory courses in high school [14], participating in out-of-school programing [41, 42], and having strong women STEM role models [43–45]. A multi-year research experience for high school girls simultaneously incorporates a number of these positive influences.

## Experiential programming

The HB SREP experience gives an all-female high school student population the chance to work in the laboratory of an engineer, research scientist, or clinical investigator for 2–3 years.

Students generally read and study the literature in the field and learn the relevant techniques while assisting others in the lab before taking on an original research project of their own. This is usually a small piece of the overarching research goals of the lab. The multi-year commitment allows the student time to develop and establish confidence in herself and her abilities, bridging the confidence gap that is often cited as one facet of why women leave the STEM pipeline.

At the time the SREP was established, internships were largely reserved for undergraduates or non-college track students in secondary schools with a focus on vocational training [46]. This made the SREP unique with its focus on college-prep, but similar to an internship in that the experience in science research or engineering was meant to expose participants to the job, gain career-based skills, and learn the professional environment [47, 48]. Generally, STEM internships are most often associated with the undergraduate phase; however, high school students can also benefit from this early career exposure and participation in research. In fact, the skills and confidence that are borne from a science or engineering experience may be one of the best recruitment strategies to increase the number of women and women leaders in STEM fields. The benefits of these experiences tend to directly counteract reasons women leave these disciplines, particularly a lack of confidence lodged in perceived incompetence [49] as well as providing an out-of-school experience [41, 42] and exposing students to more women role models in STEM occupations [42–45]. Anecdotally, the SREP was initiated when it was recognized that most elementary and middle school students were empowered in STEM, but opportunities that reinforced those feelings were few in high school resulting in many females losing interest and confidence in STEM. The SREP was created to meet this need by giving students the opportunity to continue exploring STEM and prevented their early departure from the pipeline [30, 41].

## Purpose of the study

Here, we take a retrospective view of 20 years of SREP alumnae (1998–2018) to determine if participation in the SREP increased retention in the STEM pipeline and whether outcomes were attributed to the bridging of the confidence gap through SREP during the critical high school years. Pipeline indicators were measured by declaration of STEM college majors, completion of those majors, and occupational outcomes. HB SREP participants were compared with HB students who did not participate in SREP in the same timeframe (HB Non-SREP), students nationally, and White and Asian female subsets of the national data as these particular race/ethnic groups are a better match to demographics of SREP participants. We also examined the completion of professional and graduate degrees by SREP alumnae in relation to the same groups to reflect on qualifications for leadership. Finally, the perceived traits and skills, particularly confidence and competence, acquired during the SREP experience and reported by SREP alumnae are explored as we believe that addressing the confidence gap through this model positively affects pipeline outcomes and could warrant the implementation of programs like the SREP elsewhere to bolster women in STEM fields.

Given the 20 years of alumnae data on a program housed in an all-female high school environment coupled with a multi-year experience that cultivates the relationship of the student in a professional science setting, we found ourselves in the unique position to evaluate the impact of these factors. We present evidence that women who have participated in the SREP are more likely to pursue a STEM major and occupation compared to both national norms and specific subgroups and alumnae credit their SREP experience with strengthening their confidence and empowerment, technical and professional skill set, and giving them the voice to be heard as women in their post high school careers.

The aim of this retrospective study was to gain an understanding of whether an internship-like experience in high school in a STEM field can influence retention of women in the STEM pipeline and whether this is attributed in part to participation in the SREP.

## Methods

### Ethics statement

This study was deemed exempt by the Hathaway Brown School Institutional Review Board as the research was conducted in a commonly accepted educational setting involving normal educational practices on existing data. All sources were de-identified after merging.

### Program description

The SREP is a program at Hathaway Brown School, an all-female private school for K-12 in Shaker Heights, OH that partners with academic, industrial, and private institutions to promote research experiences for students in grades 9–12. Students interested in the experience can learn about the program in grade 9 or 10 through a one semester Introduction to Research Seminar class that meets once per week. Students range from those interested in science and engineering to those unsure what a career in one of these fields might entail and are simply enrolling for exploration. In class, students learn about research areas through discussions with the Director of Research, listen to presentations from older SREP students on their projects, and read research spotlights in scientific journals in addition to learning how to write cover letters, resumes, and communicate effectively. Through these processes, students can decide if they wish to pursue a placement and if so, can focus in on particular areas and eventually begin identifying faculty at local institutions with whom they might like to work by reading institutional webpages. Students interested in pursuing a placement with a research group follow the Introductory class with a Research Seminar class that meets once per week every semester until graduation. Curriculum within the Research Seminar class includes learning how to locate and read literature to bolster scientific literacy, practice sharing work through poster and oral presentations to hone presentation skills, and culminates in the writing of a formal paper written in the format of a scientific research article.

To obtain a placement with a research group arranged by the Director of Research students must have As and Bs in their classes, receive recommendations from their teachers, meet standardized test score requirements (which have changed over time reflecting changes in administered tests but generally include 60% of the HB 9–12 student population), and commit to volunteering 2–3 summers in the lab for seven weeks of full days and once per week after school during the academic year. Students apply by submitting a cover letter, resume, and list of scientists who interest them to the Director of Research, who then reaches out on their behalf to inquire about a placement. The Director of Research individually matches students with a laboratory aligned with the student's personal interests by contacting the principal investigator, research lead, or manager directly and explaining the program, particular student, and goals of the SREP. Students who cannot fulfill the academic or time requirements or simply wish to have more flexibility may endeavor to find a placement through family connections and are still supported by the Director of Research and Research Seminar class.

The Director of Research initiates a meeting and interview with the student and principal investigator or lead and their group after mutual interest is determined and then the student is encouraged to navigate her placement independently with the Director of Research available for support and coaching. Students develop communication and organizational skills while they form a relationship with the research group. The student is typically paired with a direct supervisor like a postdoctoral scholar, graduate student, or technician in the lab for mentoring

on a day-to-day basis and learns the techniques and specialized information relevant in order to eventually earn the privilege to work on an original research project of her own that aligns with the lab's overarching goals and will contribute to the research of the group. The necessity of planning schedules and working through conflicts in addition to the nature of research and inherent setbacks develops confidence, courage, empowerment, reliability, and responsibility. It is likely that these soft skills, in addition to the science and engineering core, provide the structure to succeed further down the pipeline.

From 1998–2018, SREP students ($n = 495$) completed 619 projects that were categorized into project areas according to standard science and engineering disciplines. Some students participated in more than one project in the same lab and a trivial number completed projects in different labs. Every project was classified categorically by discipline. During the first 20 years of the SREP, biological and life science projects were the most prevalent (39%). The second highest percentage of students participated in engineering projects (30%) including materials science, macromolecular, chemical, biomedical, robotics, and mechanical and aerospace engineering. The remaining project disciplines are substantially lower, with clinical and bioinformatics at 14%, physical sciences (such as chemistry, physics, and geology) at 8%, and the rest at 5% or below (Table 1).

In addition to information collected on SREP projects while attending HB, data was collected to monitor planned college majors for SREP participants.

## Sources

**Hathaway Brown School alumnae database.** The HB Advancement Office maintains a database on all HB alumnae. For the purposes of this research, data on all HB alumnae from 1998–2018 ($n = 1,718$) were obtained providing data on race/ethnicity demographics, awarded college majors, graduate and professional degrees, and occupations for both SREP ($n = 495$) and Non-SREP alumnae ($n = 1,224$). The database is not comprehensive for every data point due to the nature of self-reporting and thus sample size varies for each research question. In some cases, subsets of the 20-year group were analyzed if certain graduation years had not had time to reach the milestones analyzed (not yet completed their undergraduate education, etc.). HB does not collect information on household income, guardian educational background, or family structure, but attracts predominantly middle-to upper-class families that are likely more highly educated than most American families.

Complete records of race/ethnicity data were only available starting with the graduating class of 2012; therefore, for combined 1998–2018, 27.5% of race/ethnicity data (473 students) are not reported for all students with 19.8% (98 students) reflected within the SREP participant

**Table 1. Categorical distribution of project discipline.** Percentage of SREP projects from 1998–2018 across STEM fields.

| Discipline | % of SREP Projects |
|---|---|
| Biological and life sciences | 39% |
| Engineering | 30% |
| Clinical and bioinformatics | 14% |
| Physical sciences | 8% |
| Math and computer sciences | 2% |
| Social sciences | 2% |
| Other | 5% |

Source: Compiled SREP data, $n = 619$, 1998–2018.

**Table 2. Race/ethnicity of HB students from 1998–2018 and public secondary school students.**

| RACE/ETHNICTY | HB | HB SREP subset | Male and Female Public Nationally[†] |
|---|---|---|---|
| White | 67.9% | 63.5% | 56.8% |
| Asian | 11.9% | 23.9% | 5.0% |
| Black | 14.6% | 7.1% | 16.1% |
| Hispanic | 1.5% | 1.5% | 19.8% |
| American Indian/Alaskan Native | 0.1% | 0.0% | 1.1% |
| Multiracial or Race Not Listed | 4.0% | 4.0% | 2.2% |

Source: Compiled SREP data, 1998–2018, HB (*n = 1,246*), HB SREP (*n = 397*).

[†]Source: US Department of Education, National Center for Education Statistics, Common Core of Data, Table 203.60, "National Elementary and Secondary Enrollment by Race/Ethnicity Projection Model, 1999 through 2018", *n = 293,330,800*).

population (Table 2). Of those reporting (*n = 1246*), HB had an overrepresentation of White students (67.9% vs. 56.8% nationally; 95% CI: 65.2%, 70.5%, p<0.001) and Asian (includes Chinese, Indian, Pacific Islander and other Asian subgroups), [50]) students (11.9% vs. 5.0% nationally; 95% CI: 10.2%, 13.8%, p<0.001) compared to national public-school students graduating between 1998–2018 [50]. Representation of Black students was similar to national percentages (14.6% vs. 16.1% nationally; 95% CI: 12.7%, 16.7%, NS), but there was a significant underrepresentation of Hispanic students (1.5% vs.19.8% nationally; 95% CI: 0.9%, 2.4%, p<0.001). In the HB SREP (*n = 397*), the subpopulation of HB alumnae of interest in this study, White students were still slightly overrepresented (63.5%) compared to public students nationally (56.8%; 95% CI: 58.5%, 68.1%, p<0.05) and there was a greater overrepresentation of Asian students (23.9%) compared to both HB in general (11.9%; 95% CI: 19.9%, 28.5%, p<0.001) and to public students nationally (5.0%; 95% CI: 19.9%, 28.5%, p<0.001). Despite similar representation of Black students between overall HB alumnae and students nationally, there was an underrepresentation of Black students in the SREP (7.1%) compared to all HB alumnae (14.6%; 95% CI: 4.8%, 10.1%, p<0.001) and students nationally (16.1%; 95% CI: 4.8%, 10.1%, p<0.001) and Hispanic student's underrepresentation among all HB alumnae was similarly reflected in the SREP (1.5% vs. 19.8% nationally; 95% CI: 0.6%, 3.4%, p<0.001). American Indian and Alaskan Native students were comparable across all populations. Finally, while those students who listed Multiracial or Race not listed were higher among all HB alumnae (4.0%, 95% CI: 3.0%, 5.3%, p<0.001) and HB SREP students (4.0%, 95% CI: 2.3%, 6.6%, p<0.05) compared to national public-school students (2.2%), it is likely that this difference reflects the variation in capturing racial identity [51]. In summary, given the overrepresentation of White and Asian students within the SREP, for the purposes of this study SREP alumnae are compared to averages for females nationally as well as to both White and Asian female subsets as these subgroups generally have a higher representation in STEM and serve as a more representative comparison group of the students enrolled in the SREP [13, 52].

**2018 SREP alumnae survey.** A survey specifically targeted to alumnae of the SREP to supplement alumnae records and assess outcome (*n = 154 respondents*) was completed in 2018. All SREP alumnae with information in HB's database were contacted to participate. The questionnaire included basic demographic (name, year of HB graduation), educational (declared major, awarded major, institution, graduation year, and any graduate or professional school attendance) and occupational information, and a question categorizing SREP experience (short answer of description of location, type, and number of years). Additional questions sought to explore the impact participation in SREP might have had on these outcomes. One

**Table 3. Public sources utilized in analyses by source and comparison.**

| SOURCE | DATA COMPARED | FIG OR TABLE |
|---|---|---|
| U.S. Department of Education, National Center for Education Statistics, Common Core of Data, Table 203.60, "National Elementary and Secondary Enrollment by Race/Ethnicity Projection Model, 1999 through 2018" [50]. | Race/ethnicity | Table 2 |
| National Science Board, National Science Foundation. Science & Engineering Indicators 2016, Appendix Table 2–16, "Freshmen intending S&E major, by field, sex, race, and ethnicity: 1998–2014" [13]. | Declared major | Fig 1 |
| National Center for Education Statistics, 2011–12 Beginning Postsecondary Students Longitudinal Study First Follow-up, Table 2–12, "Major switching among first-time postsecondary students beginning 4-year colleges and universities in 2011–12: 2013–14" [53]. | Major switching | Fig 2 |
| National Center for Educational Statistics, Table C-12 (1995–2004) [54], Table 5–4 (2004–2014) [55], Table 5–4 (2006–2016) [56], "Bachelor's degrees awarded to women, by field, citizenship, ethnicity, and race" for 1998–2018. | Awarded majors | Table 4 |
| IPUMS: 2015–2019 5-year American Community Surveys (ACS), Female 20-40-year-olds with high school or higher education grouped by race/ethnicity [52] | Educational attainment | Fig 3 |
| IPUMS: 2015–2019 5-year ACS, employed 20-40-year-olds with high school or higher education grouped by occupation, sex, and race/ethnicity [52] | STEM occupations | Fig 3 |

multiple answer question assessed attributes or skills respondents felt were strengthened from participation in the SREP (confidence, courage, empowerment, professionalism, reliability, responsibility, communication skills, critical thinking, presentation skills, and scientific literacy) (*n = 152*). Two optional, open-ended short answer questions sought responses regarding the role the SREP experience played in character and skill development (*n = 141*) and how being female has affected alumnae careers (*n = 114*). The most common categories of responses were identified and were coded and quantified accordingly.

**Other sources.** Publicly accessible databases and statistics were used for comparison information (Table 3).

## Data analysis

**Classification of degrees according to major field of study.** To classify major field of study for both declared and actual majors, taxonomies followed the broad categories assigned by the NCES breakdown of college majors [57]. A response of other and not classified with a write-in or database entries that did not clearly fit a broad category were compared to the minor categories listed by the NCES and renamed accordingly. To compare the data with the NSF data of STEM majors, health professions and related programs were considered Non-STEM majors unless otherwise indicated. The following NCES majors were grouped into NSF categories: both agriculture and natural resources as well as biological and biomedical sciences into NSF biological/agricultural sciences; engineering into NSF engineering; both computer and information science with mathematics and statistics into NSF mathematics, statistics and computer science; physical sciences and science technologies into NSF physical science; both psychology as well as social sciences into NSF social and behavioral sciences. If an alumna listed a double major only one was included in the analysis. If one major was a STEM major and one was not, the STEM major was included; otherwise, the first of the two listed was included.

**Designation of graduate and professional degrees.** Professional degrees included MD, DDS, DVM, LLB, or JD and variations. Doctoral degree includes attainment of a PhD or EdD. Degree field for Master's degrees was categorized by the classification of degrees according to the NCES field of study as listed above [57]. Only the highest degree attained for each

individual was included. Those earning an MD/PhD were classified under the PhD category, except when percentages were listed for individual types of degrees and the MD/PhD was calculated separately and therefore not included in either individual MD or PhD counts.

**STEM occupations.** STEM occupations include those employed in computer and mathematics, architecture and engineering, and life and physical science occupations, as well as managerial and postsecondary teaching occupations and sales occupations requiring scientific/technical knowledge according to the US Bureau of Labor Statistics [58].

## Statistical analysis and robustness of results

The one sample Z proportion test with 95% confidence intervals (95% CI) was used to compare SREP alumnae to HB Non-SREP alumnae and populations reported by the NCES or US Census Bureau. While this research approximates age and gender-matching as it focuses on age-specific groups in a female-only population, it does not account for socio-economic status, household educational attainment, or geographical location as these data were not available in the pre-existing data sets. SREP alumnae are not a representative sample of all females, although here they are compared to national female or male populations but also with HB Non-SREP students when possible as well as subsets of White female and Asian female populations as these are the predominant race/ethnicities represented within the HB SREP.

## Results

### SREP alumnae are more likely to declare a major in a STEM field than their peers

Of SREP participants, the majority of SREP alumnae (62.2%, CI: 55.9%, 68.0%) reported that they planned to major in a STEM field after they graduated high school and as they entered college (Fig 1A). This was significantly higher than White females nationally (26.8%) who

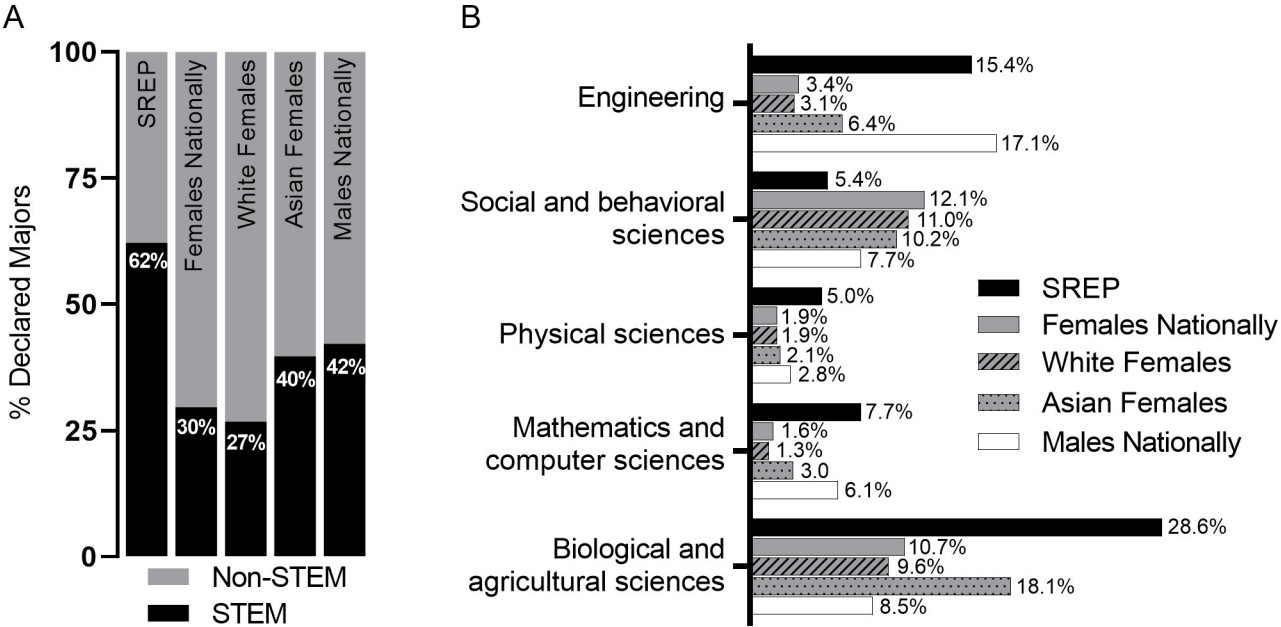

**Fig 1. Students declaring STEM majors.** (A) Percentage of SREP students from 1998–2018 (*n = 259*) declaring STEM majors compared to national averages from 1998–2014 and (B) those declaring STEM majors subdivided into disciplines. National percentages were obtained from Appendix Table 2–16 from [13], with sample size unavailable). Limiting SREP alumnae to the years 1998–2014 to match the national data does not significantly change any category.

declare STEM majors (95% CI: 55.9%, 68.0%, p<0.001) as well as Asian females nationally (39.7%; 95% CI: 55.9%, 68.0%, p<0.001). Thus, it is unsurprising that SREP alumnae declare STEM majors at a higher rate than all females nationally (29.6%) as they enter college (Fig 1A). Interestingly, SREP participants also significantly surpass the 42.2% of males nationally who declare a STEM major (95% CI: 55.9%, 68.0%, p<0.001).

The NSF defines a STEM major or career to include biological, agricultural, physical, computer, and mathematical sciences; engineering; social sciences; and psychology, but excludes the health sciences. Only 5.4% of SREP students declare a social or behavioral science major as they are graduating, significantly less than the 12.1% national female average (95% CI: 3.1%, 9.1%, p>0.001) or White (11.0%, 95% CI: 3.1%, 9.1%, p>0.01) and Asian (10.2%, 95% CI: 3.1%, 9.1%, p>0.05) subgroups. Instead, SREP alumnae most often declare a major in the biological and agricultural sciences (28.6%), specifically the biological sciences as there were no responses from the agricultural sciences (Fig 1B). Nationally, Asian females choose biological and agricultural science majors as the most popular STEM major (18.1%), but at rates lower than SREP participants (95% CI: 23.2%, 34.5%, p>0.001). White females (9.6%) choose this category in similar percentages to all females nationally (10.7%) on par and slightly higher than males (8.5%; Fig 1B). The interest and representation in the biological sciences of SREP alumnae is consistent with the large number of projects within this discipline in SREP (Table 1) and increases representation in this discipline significantly above all other groups compared (Fig 1B).

The percentage of SREP alumnae reporting declared majors in mathematics/computer science, physical sciences, and engineering was greater than the national female average, but also significantly higher than White and Asian female subsets and reflected the breakdown of projects students experienced while in the SREP (Table 1). This difference was significant in all three discipline groups compared to Asian females, who had the highest percentages of majors declared in each of these fields of females overall. 7.7% of SREP students declared a major in mathematics and computer science compared to Asian females nationally at 3.0% (95% CI: 4.9%, 11.8%, p<0.001), 5.0% of SREP students declared a major in the physical sciences (compared to 2.1% for Asian females nationally; 95% CI: 2.8%, 8.6%, p<0.01), and 15.4% in engineering, which was well above the national female average at 3.4% (95% CI: 11.4%, 20.6%, p<0.001), and still significantly above the 6.4% Asian females who nationally declare an engineering major (95% CI: 11.4%, 20.6%, p<0.001; Fig 1B). Interestingly, the percentage of SREP students planning to major in each of these STEM disciplines that have been historically male-dominated was on par with their male counterparts (Fig 1B).

## SREP graduates are less likely to switch from a STEM major to a Non-STEM major and are awarded more STEM bachelor's degrees than comparison groups

SREP student graduates from 1998–2014 with recorded declared majors and completed degree information (*n = 238*) were assessed for switching between STEM and Non-STEM majors between their declared major and awarded major and compared to NCES records of combined male and female first-time postsecondary students from 2011–2012 and their reported majors in 2013–2014. These reported majors from NCES were collected from the same students two years after they began their post-secondary education and not necessarily their completed major as with the SREP alumnae [53]. Over two-thirds (76.9%) of SREP graduates selected a STEM major, over twice the number of students nationally (32.9%), which includes both males and females (95% CI: 70.9%, 82.0%, p<0.001; Fig 2A). Interestingly, after at least two years, this large gap was expanded, as SREP graduates more often persisted with a STEM

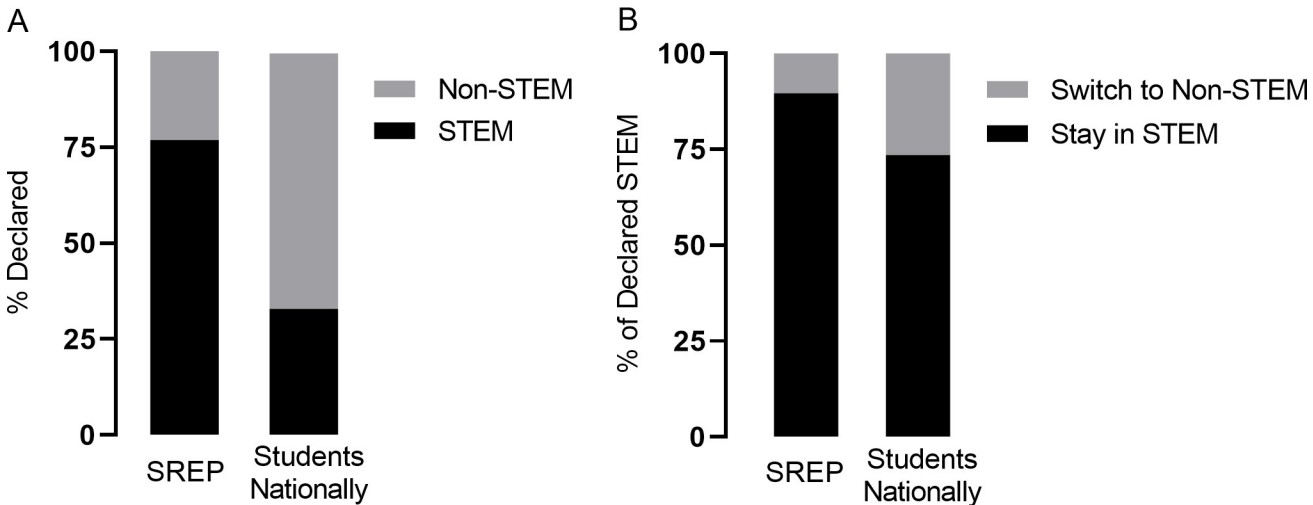

**Fig 2. SREP graduates declare and persist in STEM majors more often than students nationally.** Percentage of students (A) declaring a STEM major when entering college out of all majors and (B) staying in a STEM major out of those who originally declared a STEM major. SREP students declared and actual from 1998–2014 (*n = 238*) compared to 2011–12 male and female freshman assessed again in 2012–13 (*n = 2,237,000*, Table 2–12 from [59]).

major (89.6%), compared to students nationally (73.6%; 95% CI: 84.0%, 93.5%, p<0.001; Fig 2B). Students transferring from Non-STEM to a STEM major was comparable across both groups (23.6% of SREP graduates compared to 14.5% of students nationally; 95% CI: 13.7%, 37.3%, NS).

SREP alumnae persist with their major more often than females nationally (Fig 2B). To determine if the all-female SREP alumnae earn STEM degrees at higher rates than similar groups, all SREP alumnae with reported majors graduating HB between 1998–2016 were included (*n = 288*) and almost two-thirds (61.5%) of bachelor's degrees earned by SREP alumnae were in STEM fields (Table 4). Nationally, data averaged from 1998–2016 revealed that a little over a quarter of bachelor's degrees during this time frame were awarded to females in STEM fields (28.4%; 95% CI: 55.5%, 67.1%, p<0.001). Rates of White females awarded STEM degrees were similar to all females (26.5%). Asian females were awarded degrees in STEM fields at higher rates than the female average, but still at significantly lower percentages than

**Table 4. Females awarded bachelor's degrees in STEM fields.**

| FIELD | % Bachelor's degrees awarded | | | | |
|---|---|---|---|---|---|
| | SREP* | Non-SREP* | All Females Nationally† | White Females† | Asian Females† |
| **Biological/agricultural sciences** | 23.5% | 3.6% | 6.8% | 6.6% | 13.6% |
| **Engineering** | 11.9% | 2.9% | 1.6% | 1.4% | 3.7% |
| **Mathematics/computer sciences** | 7.7% | 2.5% | 1.9% | 1.6% | 3.7% |
| **Physical sciences** | 4.6% | 2.2% | 1.0% | 1.0% | 1.7% |
| **Social/behavior sciences** | 14.4% | 24.2% | 17.0% | 15.9% | 19.2% |
| **STEM combined** | 61.5% | 35.4% | 28.4% | 26.5% | 41.9% |

*Source: HB Alumnae Database: (1998–2016), SREP, *n = 288*; Non-SREP, *n = 277*

†Source: National Center for Educational Statistics, Table C-12 (1995–2004) [54], Table 5–4 (2004–2014) [55], Table 5–4 (2006–2016) [56], "Bachelor's degrees awarded to women, by field, citizenship, ethnicity, and race". Averaged across 1998–2018; no data for 1999. All females, (*n = 16,226,416*); White females (*n = 10,455,689*), Asian/ Pacific Islander females (*n = 973,331*).

Physical sciences = chemistry, physics, astronomy, and earth/atmospheric/ocean sciences.

SREP alumnae (41.9%; 95% CI: 55.5%, 67.1%, p<0.001). HB Non-SREP students from the same graduation years with reported awarded degrees were also compared (*n = 277*). Data from this group controls for socioeconomic status, family structure, and parental education to a certain degree as students attend the same school, but without the SREP experience. HB Non-SREP students major in STEM fields at a rate (35.4%) that falls between all females nationally and the Asian female subgroup but at a significantly lower rate than their SREP counterparts (95% CI: 55.5%, 67.1%, p<0.001).

The increase in awarded STEM degrees among SREP alumnae occurs most prominently in disciplines that are less common for females, such as mathematics and engineering. The rate at which SREP alumnae earn degrees in these disciplines reaches significance even when compared to the female subgroups with the highest rate of degrees in each discipline. For example, 7.7% of SREP alumnae major in mathematics and computer science compared to 3.7% of Asian females nationally (95% CI: 4.9%, 11.5%, p<0.001). A physical sciences major is attained by 4.6% of SREP alumnae compared to 2.2% for HB Non-SREP alumnae (95% CI: 2.5%, 7.8%, p<0.01), and 11.9% of SREP alumnae major in engineering compared to 3.7% of Asian females nationally (95% CI: 8.4%, 16.2%, p<0.001).

### Students with experience in the SREP are more likely to complete graduate or professional school and choose a STEM occupation

SREP alumnae's pursuit of advanced degrees and professional school was assessed to determine if participation in SREP might influence further education which also equips the individual for additional career advancement and leadership roles. All SREP alumnae graduating between 1998–2013 (*n = 336*) were included for analysis. These data were compared to publicly available census data on 20-40-year-old females who had completed high school or higher education. SREP alumnae completed significantly more Master's degrees (18.5%) than the average of all females in this dataset (*n = 41,883,102;* 9.3%; 95% CI: 14.5%, 23.1%, p<0.001; Fig 3A); however, they did not statistically differ when compared to only the Asian female subset (*n = 3,053,851;* 18.3%; 95% CI: 14.5%, 23.1%, NS). HB SREP alumnae also completed twice as

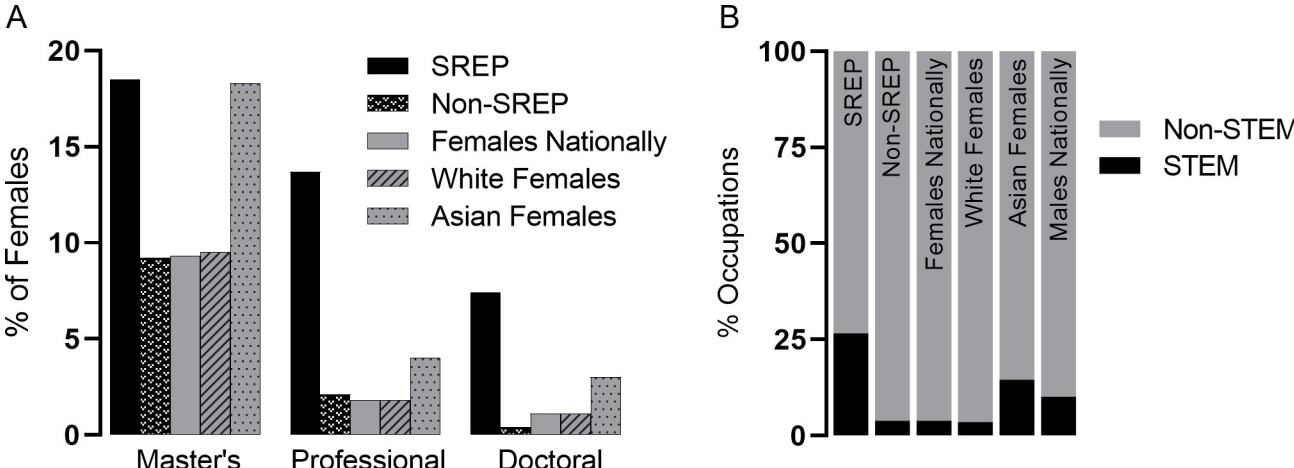

**Fig 3. Post-graduate education and STEM occupation are impacted by SREP participation.** (A) Percentage who completed a graduate or professional degree (including MD, DO, DDS, DVM, LLB, or JD) of SREP alumnae from 1998–2015 (*n = 336*) compared to 1998–2013 HB Non-SREP alumnae (*n = 915*) and all female (*n = 41,883,102*), White female (*n = 28,947,767*), and Asian female (*n = 3,053,851*) 20-40-year-olds with a high school education or above from the 2015–2019 5-year ACS [52]. (B) Percentages of STEM and Non-STEM occupations for SREP alumnae from 1998–2013 reporting (*n = 286*) compared to 1998–2013 HB Non-SREP alumnae reporting (*n = 560*) and all employed female (*n = 31,297,104*), male (*n = 33,955,393*), White female (*n = 21,962,960*), and Asian female (*n = 533,709*), 20-40-year-olds with a high school education or above from the 2015–2019 5-year ACS [52].

many Master's degrees as their HB Non-SREP counterparts (*n = 915;* 9.2%; 95% CI: 14.5%, 23.1%, p<0.01). For SREP alumnae, Master's degrees were distributed as 58.1% Non-STEM, 29.0% biological or medical, and 8.1% engineering, mathematics, computer and physical sciences. Additionally, students with SREP experience were more likely to pursue a professional degree compared to HB Non-SREP students (13.7% SREP vs 2.1% HB Non-SREP; 95% CI: 14.5%, 23.1%, p<0.001) and compared to Asian females, the subgroup with the highest percentage in the dataset (4.0%; 95% CI: 14.5%, 23.1%, p<0.001). Lastly, 7.4 of SREP alumnae have earned a doctorate, significantly more than other females their age (1.1%; 95% CI: 5.0%, 10.9%, p<0.001), than HB Non-SREP alumnae (0.4%; 95% CI: 5.0%, 10.9%, p<0.001), and twice the rate of the Asian female subgroup (3.0%; 95% CI: 5.0%, 10.9%, p<0.001; Fig 3A). Professional and graduate degrees pursued by SREP alumnae included JD (3.9%), MD (7.4%), PhD (6.0%), MD/PhD (1.2%), DDS (0.9%), and DVM (0.9%).

To determine if SREP alumnae are more likely to continue in the STEM pipeline and become employed in a STEM occupation, SREP alumnae with reported occupations who graduated between 1998–2015 (*n = 286*) were compared to census data on age-matched controls of both sexes [52] as well as to HB Non-SREP alumnae (*n = 560*). Of those SREP alumnae with reported occupations, 26.6% were in a STEM occupation compared to 3.9% of HB Non-SREP alumnae (*n = 560*; 95% CI: 21.5%, 31.7%, p<0.001, Fig 3B). Nationally, Asian females from the census dataset (*n = 2,060,227*) had the highest percentage engaged in STEM occupations among females (13.2%, Fig 3B), but HB SREP participants were still twice as likely to persevere in the STEM pipeline by choosing a STEM occupation (95% CI: 21.5%, 31.7%, p<0.001). SREP alumnae also significantly surpassed the 10.1% of males in this age range with high school degrees or above that are employed in STEM occupations (*n = 33,955,393;* 95% CI: 21.5%, 31.7%, p<0.001; Fig 3B), indicating that participation in the SREP could potentially contribute to STEM pipeline retention.

### SREP alumnae believe their experience strengthens critical character traits and teaches skills that prepare them for future careers

In a 2018 survey, SREP alumnae were asked to mark all character traits and skills that applied to the question "which of the following attributes or choices were strengthened by your participation in the SREP" with the potential answers including the attitudinal goals of confidence, courage, empowerment, professionalism, reliability, and responsibility (Fig 4A) as well as content goals of communication skills, critical thinking, presentation skills, and scientific literacy (Fig 4B). Interestingly, all categories but one was selected by over half of respondents (*n = 152*) acknowledging the impact the SREP played in strengthening necessary skills and attributes. In

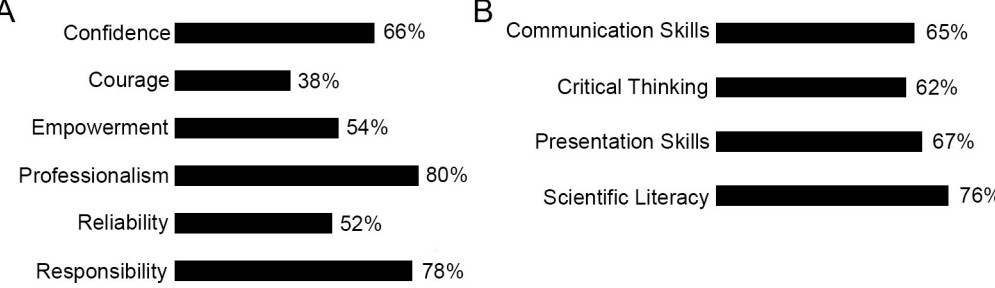

**Fig 4. SREP alumnae credit their SREP experience with the strengthening of critical personal attributes and skills.** Respondents (*n = 152*) indicated which (A) attitudinal goals and (B) content skills were increased through their participation in the SREP.

particular, over three quarters of students felt that their experience bolstered their professionalism (80%) and responsibility (78%) and two-thirds cited a building up of confidence (66%; Fig 4A). Close to two-thirds of all respondents also selected all skill categories with scientific literacy at the highest (76%) and presentation skills (67%) close behind (Fig 4B).

To gain more insight into the attribute and skill categories, SREP alumnae were also asked two open ended questions. These questions did not have selection options and all responses were read and coded for overarching categories initiated by the respondents themselves. Responses often fell into more than one category. The first question was "How did the SREP play a role in your education, career, or personal development?" with 141 responses, a 92% response rate of the 154 alumnae that completed the survey in 2018. The largest trends in responses included the learning of broad critical professional skills and/or scientific technical skills that led to a feeling of being well-prepared for the later stages of their careers (45%) and a confirmation of interest in STEM or health science careers (41%). Of their own accord, 23% of respondents mentioned confidence—whether in themselves, their qualifications, or their ability to successfully pursue their goals (Fig 5A).

A second open ended question, "Would you say that being female has affected your post-HB career? In what ways—positive or negative?" garnered a 74% response rate (*n = 114*) of the 154 total respondents graduating between the years of 1998–2017. The most common answer included acquiring a "strong voice" or being empowered as a woman though participation in SREP or by attending HB (29%). SREP alumnae also frequently reported that they were under more scrutiny or more pressure to prove themselves than their male counterparts (27%) and that family responsibilities affected them more often as women (15%; Fig 5B).

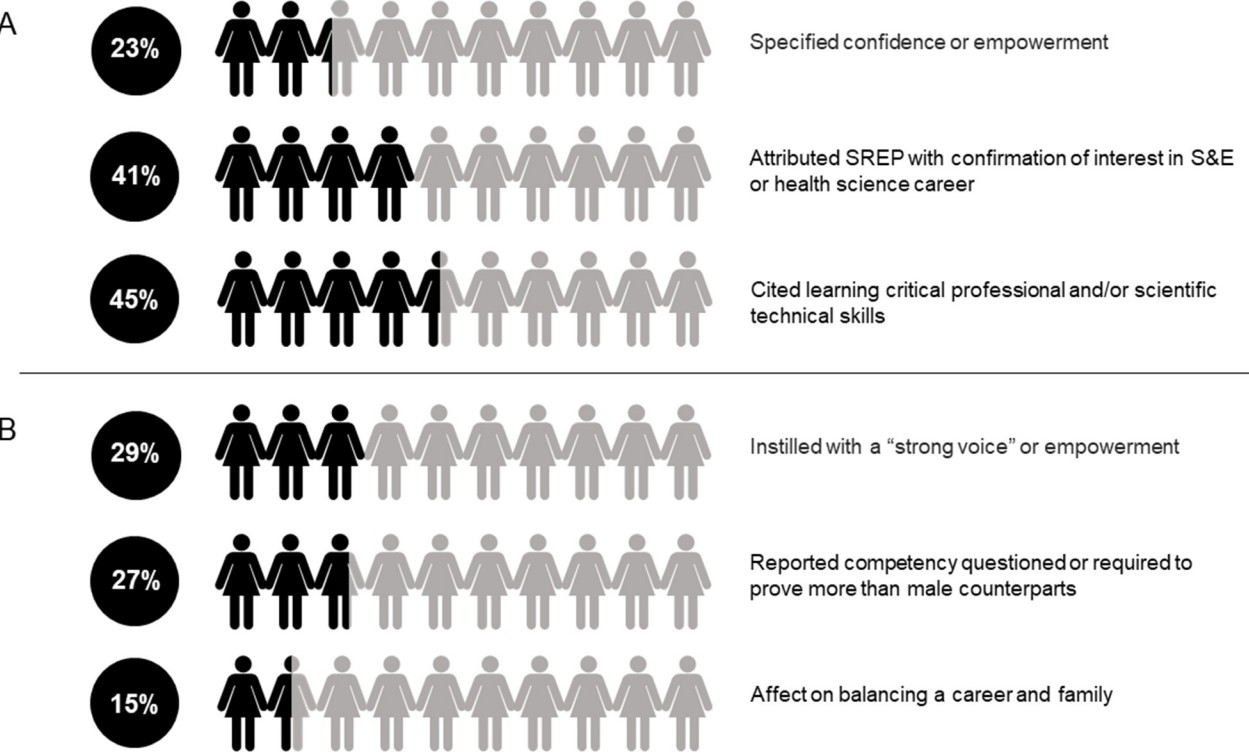

**Fig 5. SREP alumnae are instilled with confidence and empowerment to persevere in the STEM pipeline.** Percentage of respondents answering by theme on (A) the impact of the SREP (*n = 141*) and (B) of being a woman in their post-HB career (*n = 114*).

## Discussion

The aim of this study was to determine if participation in an internship-like experience in a STEM field through the SREP at an all-female school influenced perseverance in the STEM pipeline. Based on the data presented here, it appears the SREP is successful to this goal. SREP alumnae more often declare a STEM major than comparison groups (Fig 1) although there were variations within disciplines. Nationally, the social or behavioral sciences majors have the highest representation of women within STEM majors (17%), and while a smaller percentage of SREP students declare a major in this field (6.5%), a similar number of SREP alumnae change majors and earn a bachelor's degree in the social and behavioral sciences (17.3%; Table 4). The SREP has historically focused on the basic sciences and social sciences have not been highly represented within the SREP (2% of projects; Table 1). Because of this, students may be enthusiastic about a STEM major based on their SREP experience, but then move horizontally across STEM disciplines as they are exposed to new experiences and coursework.

The beginning of high school is known to be a critical time in the STEM pipeline [29] and is when students begin in the SREP. Thus, a positive experience and confidence in ability during this critical period could potentially steer students in pursuit of STEM majors and occupations and prevent girls who are interested in STEM from shifting away from STEM in college or later. By and large, the biological sciences are the primary declared major for SREP alumnae (Fig 1). There is high representation in biological and life science projects within the SREP (39%; Table 1), anecdotally because many SREP students have the end goal of an advanced degree in medicine with a pathway through the biological sciences. Unfortunately, this cohort is not captured well in the data reported here because the health professions and related fields major is not considered a STEM major. This is likely because most students nationally who major in health professions become technicians and nurses rather than medical doctors; however, many SREP alumnae majoring in this discipline go on to obtain a professional degree (Fig 3A). The biological and agricultural sciences major is also a common route for medical professionals and is highly selected by SREP alumnae, but certainly the percentage of students choosing a STEM major omits those SREP alumnae who choose to major in health professions and related fields (12%) and went on to pursue an advanced degree.

Attrition rates in college for STEM majors are often linked to grades, and women in particular are more likely to change their major due to poor performance rather than continuing on with a few lower grades than their male counterparts in male-dominated fields [16]. In addition to declaring a STEM major at higher rates (Figs 1 and 2A), SREP students persist in following through with a STEM major more often than their national counterparts (Fig 2B, Table 4). This may be because students participating in the SREP have the advantage of experimenting with a STEM profession before pursuing it fully. The experience allows students to determine if a research career is of interest, but additionally gives them the resources, relationships, and networks to explore additional occupations in STEM fields before committing [42, 60]. While SREP alumnae have strong representation in all STEM fields, the increase in the fields of mathematics and computer science, physical sciences, and engineering are especially encouraging. These particular STEM disciplines are of critical focus as women have earned a smaller percentage of these degrees (mathematics/computer science at 33%, physical science at 40.5%, and engineering at 19.6%) over the last 18 years [11,12] and concerns of confidence and perceived competence have often been raised as potential barriers most acutely in these fields [19, 20, 23, 26]. With 61.5% of SREP alumnae majoring in STEM fields, it might appear that the SREP self-selects for STEM-minded students. However, 35.4% of HB Non-SREP students major in a STEM field (Table 4), above the national female average, which suggests the SREP experience enriches STEM outcomes overall.

After finishing their undergraduate degree, SREP alumnae go on to pursue graduate and professional school at higher rates than their HB Non-SREP counterparts and most female subgroups (39; Fig 3A). Importantly, while not all of these post-graduate studies are within STEM fields, they indicate attainment of higher qualifications for women, positioning them for potential leadership roles across their chosen field. Given the lack of women in senior positions across fields, this alone is a noteworthy accomplishment [27, 61]. Considering the sample included all SREP alumnae from 1998–2013, even those without updated records, it is likely that if all data were available, completed records would reveal an even greater increase from controls. The combination of post-graduate degrees coupled with a strong representation in undergraduate STEM degrees (Table 4) translates well to the higher number of SREP alumnae in STEM occupations. Not only does the percentage of SREP alumnae in STEM occupations well exceed those of women nationally, it also exceeds those of their male counterparts (Fig 3B). While other variables are certainly at play, the experience of SREP seems to strongly influence alumnae toward STEM occupations.

Participating in SREP gives students the chance to act autonomously, without teacher or parent oversight, while developing collegial relationships with adults. This learning structure has been shown to be one of the best predictors of personal growth experiences [42, 62]. Students also develop additional mentors in professional fields and directly see female role models in their anticipated career fields, opportunities that have been shown to positively influence women toward STEM majors [36, 37, 44, 45]. It could be that for SREP alumnae it is a coupling of an impactful experience that concurrently builds confidence and captures an interest in STEM fields at the critical high school years [30, 42]. Studies demonstrate that the initial choice of a STEM major is impacted by confidence and sex disparities in this critical character trait contribute to the gender gap in these fields [26]. Through their time in research, SREP students develop confidence in their abilities in the scientific arena and define themselves within a professional environment (Figs 4 and 5) [14, 21]. Whereas the national female averages in selecting STEM majors (Table 4) and ultimately occupations (Fig 3B) demonstrates how gender differences in self-belief, including math self-efficacy and fear of failure, still largely favor males [1], women who have participated in the SREP self-report they have overcome some of these obstacles even though they face many of the same societal issues reported by women nationally (Fig 5) [4, 63].

## Limitations

Students elect to enroll in the SREP during grade 9 or 10 in high school. Anecdotally, students who sign up are those already interested in STEM fields as well as those who are using the SREP to explore their options. However, it is likely that a selection bias exists in that students who have enrolled in SREP are already interested in STEM fields and would have had a higher probability of pursuing this track. This is controlled to a certain degree by the inclusion of data on HB Non-SREP students who, even excluding the SREP students, seem to perform on par with national female averages (Table 4, Fig 3) indicating that not all students who pursue STEM participate in the SREP but that the increase in STEM perseverance is specific to the program. Similarly, many SREP students have reported that their experience actually drove them away from STEM fields as it provided clarity on their own personal interests (2018 SREP Survey; 20%; *n = 152*); therefore, we believe it is still a reasonable and valuable comparison. HB students in general come from middle to upper class families with higher educational attainment and thus are not representative of the general female population. Finally, parameters for this analysis were dependent on self-reporting, whether through survey or through contact with the HB alumnae office and this in itself lends the data to reporting bias.

## Conclusion

Assessment of outcome of educational programs is critical to determine if the program is working as planned and meeting the goals it set out to accomplish. If so, similar programs are likely to have comparable achievements. We have shown that students who participate in the SREP and have a high school research experience are more likely to pursue a STEM degree and occupation as measured against Non-SREP counterparts, national averages of males and females, and particular female subgroups. Especially notable is the increase in awarded STEM degrees to SREP alumnae within STEM disciplines that are traditionally male-dominated and suffer from effects of the confidence gap, such as mathematics and engineering, which supports the hypothesis that that it is an increase in confidence that encourages these students to persist [1, 26]. The majority of students, regardless of career path, have gained confidence and skills which have better prepared them for future experiences. While additional barriers to increasing the number of women in STEM fields continue to persist, investing in the experiential opportunities of our young women in STEM during high school may be an integral step in the process of instilling confidence and engagement during this critical time period.

## Acknowledgments

We would like to thank Bill Christ, the Head of School at Hathaway Brown School when the Science Research & Engineering Program was instituted and for its first 18 years, as well as Dr. Fran Bisselle for her continued support of the program as the current Head of School. We would also like to recognize Louise Scott, Shelley Johns, and Mary Toth for managing and curating the databases used in this analysis. We are grateful to Dr. Kerry O'Connor and Dr. Marisa Macnaughtan for editorial comments. Lastly, we are incredibly grateful to the scientists, engineers, and other mentors of SREP students without whom this program would not be possible.

## Author Contributions

**Conceptualization:** Patricia K. Hunt, Crystal M. Miller.

**Data curation:** Patricia K. Hunt, Michelle Dong, Crystal M. Miller.

**Formal analysis:** Crystal M. Miller.

**Investigation:** Patricia K. Hunt.

**Methodology:** Patricia K. Hunt, Crystal M. Miller.

**Project administration:** Crystal M. Miller.

**Resources:** Patricia K. Hunt, Crystal M. Miller.

**Software:** Michelle Dong.

**Supervision:** Crystal M. Miller.

**Validation:** Crystal M. Miller.

**Writing – original draft:** Crystal M. Miller.

**Writing – review & editing:** Patricia K. Hunt, Michelle Dong.

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
