## [Decision Letter · Decision Letter 0]

14 Dec 2020

PONE-D-20-34867

A multi-year science research or engineering experience in high school gives women the confidence for future S&E occupations and advancement: A 20-year longitudinal study

PLOS ONE

Dear Dr. Miller,

Thank you for submitting your manuscript to PLOS ONE. After careful consideration, we feel that it has merit but does not fully meet PLOS ONE’s publication criteria as it currently stands. Therefore, we invite you to submit a revised version of the manuscript that addresses the points raised during the review process.

You need to address the issues regarding the statistical analysis of the data and the presentation of the results. It is important that you make sure that there is clarity in how they are presented and that this fits with your hypothesis.

We look forward to receiving your revised manuscript.

Kind regards,

Andrew R. Dalby, PhD

Academic Editor

PLOS ONE

Journal Requirements:

Reviewers' comments:

Reviewer's Responses to Questions

**Comments to the Author**

1. Is the manuscript technically sound, and do the data support the conclusions?

Reviewer #1: Partly

Reviewer #2: No

2. Has the statistical analysis been performed appropriately and rigorously? 

Reviewer #1: N/A

Reviewer #2: No

3. Have the authors made all data underlying the findings in their manuscript fully available?

Reviewer #1: No

Reviewer #2: Yes

4. Is the manuscript presented in an intelligible fashion and written in standard English?

Reviewer #1: Yes

Reviewer #2: Yes

5. Review Comments to the Author

Reviewer #1: It is impressive result the SREP graduates are awarded more S&E degree than the national female average. The result looks clear and teachers’ long-term efforts is wonderful, but there are several lack of description and analysis.

1.First, more data of SREP at the school is needed. There are 619 SREP projects, at least, the author should classify and add projects’ field % like Table 2. The SREP graduates ration in the “Biological/agricultural sciences” and “Engineering” are awarded more than the average, it should be discussed the relation between the project fields.

2.Not only the projects’ field, but also the more detail about the SREP program should be discussed. The actual situation cannot be seen, such as how the teachers prepared the project, negotiated, and how much they were involved in the operation.

3. The discussion is needed how the options in Fig 5 were selected. These items should be discussed in combination with the SREP design concept.

4.Overall results and discussions are not clearly separated. Especially since it is a treatise and not an article, it is not suitable to write a conclusion in the subsection title.

Reviewer #2: Lines 44-45: should say: while men and women students… equal numbers…

Line 56: is it a private school or charter school? I am not sure what an “independent school” is?

Line 80-81: If they have comparable aspirations, why would girls reference careers requiring college and boys would not? This is not clear.

Line 82: This also seems contradictory: if scores are equal, how could they favor boys? There is obviously a lot of nuance in this literature and I think the precision of the writing needs to be improved in this section of the paper.

Lines 190-200: SREP sounds amazing. I think we need more details about how the program fit within their high school coursework (during the week, did they go to the lab during school day?) In the summer, were they paid or just volunteers?

More importantly (in terms of a selection bias issue): How were students selected for it?

What are the sociodemographics of the student body at this school and in SREP? (Line 274 is too late for this info to appear. The info is also vague there—report actual percentages).

I think the authors should pose research questions at the end of the introduction. Without doing that, it is hard to see how the different data sources and variables are going to be used (on pages 9-11). For example, I don’t know how supplemental survey which 25% of the cases responded too will be used.

In the data section, the authors also do not introduce how they will determine the control group or control population. Line 263 mentions some sources, but not specifically what they are and how they are used.

Line 261: The statistics section does not discuss any actual statistical approaches. What tests are used?

Line 268: The authors report that their alumni are not nationally representative, but they need to actually show the reader how they diverge from national stats so that we can use that information to evaluate the comparisons. So compare actual percentages between groups.

Line 271: says that the purpose is to assess SREP as a model. It seems very late in the paper to be stating that as the goal. Also, is that what the analyses really achieve? In lines 278-9: another goal is stated. The paper should have one goal (or a set of related goals) and it should be introduced early in the paper and then the analyses should support that goal).

Section at Line 282: This does not seem to be results, it seems like background on the program, I would move it back to the section on SREP, unless you create new research questions that include this as one of the questions.

Line 305: This heading seems to be one of the key research questions. Pose it earlier.!

Line 316: Is there a way to get these national stats for white and Asian women alone? Since we know that minority students are less likely to go into STEM careers, the national averages are lower than they would be for the group of students from this school. There might be better comparison stats than national averages when your students are not nationally representative.

Line 361: What does being “more firmly committed” mean? I would recommend just using language about switching majors in the headings for clarity.

Line 377-378: I am not convinced that that gap could not be explained by the SES and race/ethnicity differences between the students in your sample and the national averages.

I would recommend posing a series of research questions (at the end of the intro) that lead the reader through the results.

Line 568: The issues of selection bias in this paper are HUGE and since those who do STREP are not representative in any way, I am just convinced that the national stats are the right comparison group. I think they could be utilized in the paper, but I don’t think they should be the featured story since the HB alumnae are not even close to representative (this was not reported directly, but I assume this, based on what the authors reported in the paper).

One better comparison group would be HB students who did not do STREP. I don’t see why (line 582) this group could not have been made available. That does not make sense to me, given that the study is already using alumni records. (If indeed that is the case, this should be better explained.)

The paper has a really cool set of data and I like it in many ways, but it is fundamentally flawed. Research looking at this question about the impact of program participation on STEM trajectories would use propensity score matching to match each SREP student with controls, conduct multivariate stats, and then that would isolate the effect of SREP on their trajectory and you could really say that it helped and by how much it helped.

In the absence of that, it seems like making multiple comparisons is probably best. Comparing local stats for the community/city in which HB is located, comparing by race/ethnicity nationally, comparing to national averages, etc.

Comparing to national statistics only is not very meaningful, given that it appears this group is quite a bit more affluent and advantaged than the average American woman. One possibility is to look at all the stats by race, so find national stats for white woman and Asian women and compare to them. However, since those stats for the HB alumni were not reported, I cannot provide definitive advice here on how to move forward. The current approach is problematic.

The tables and figures are missing captions and notes. The reader needs to know the N of the groups and the source of the data in all of them. I also think basic tables are missing, such as descriptive stats on the HB alumni group (and how they compare to national stats, if those stats are used).

6. PLOS authors have the option to publish the peer review history of their article (what does this mean?). If published, this will include your full peer review and any attached files.

Reviewer #1: No

Reviewer #2: No

---

## [Author Response · Author response to Decision Letter 0]

9 Jun 2021

REVIEWER 1 

1. First, more data of SREP at the school is needed. There are 619 SREP projects, at least, the author should classify and add projects’ field % like table 2. The SREP graduates ration in the “biological/agricultural sciences” and “engineering” are awarded more than the average, it should be discussed the relation between the project fields.

 Figure 1 has been converted to Table 1 (Lines 270-279, Table 1) and moved to the Methods under the “Program description” subsection to classify the project field in a similar way to the majors in Table 2.

2. Not only the projects’ field, but also the more detail about the SREP program should be discussed. The actual situation cannot be seen, such as how the teachers prepared the project, negotiated, and how much they were involved in the operation.

 The authors agree that information was lacking and have expanded on the “Program description” subsection (Lines 222-279) to include the details of how this program works and expand on the information about participants, how placements were established and the role of the Research Director, students, and curriculum.

3. The discussion is needed how the options in fig 5 were selected. These items should be discussed in combination with the SREP design concept.

 The authors have added a “Purpose of the study” subsection to the Introduction to address how the perceived character qualities that students feel they have gained through their participation in the SREP have potentially impacted the outcome (Line 263-268). In addition, these particular skills sets are called out in the “Program description” (Line 229-232 & 237-240) to explain how they are incorporated as learning objectives.

4. Overall results and discussions are not clearly separated. Especially since it is a treatise and not an article, it is not suitable to write a conclusion in the subsection title. The authors appreciate this observation and have relocated or removed the sections of text in the results that were more suited for the discussion (Lines 413-420, 439-441, 456-463, 483-486, 507-509, 546-551, 583-584, 602-605). The Conclusions section is also no longer a subsection, but a main section (Line 750).

REVIEWER 2 

Lines 44-45: should say: while men and women students… equal numbers…

 Corrected. (Line 46-47)

Line 56: is it a private school or charter school? I am not sure what an “independent school” is?

 Revised. An independent school is a subcategory of private school, but private is familiar and will suffice here. (Line 59)

Line 80-81: if they have comparable aspirations, why would girls reference careers requiring college and boys would not? This is not clear.

 The authors agree that this statement and entire paragraph is problematic. After consideration, the paragraph has been removed (Lines 142-147) as it attempted to briefly address elementary and middle school stages of the STEM pipeline, but, as the reviewer rightly observes, there is nuance in the literature that cannot be adequately summarized in a short paragraph. This entire section has been rearranged to focus on the high school experience as it impacts the STEM pipeline and excludes discussion of earlier timepoints. (Lines 128-157).

Line 82: this also seems contradictory: if scores are equal, how could they favor boys? There is obviously a lot of nuance in this literature and i think the precision of the writing needs to be improved in this section of the paper.

 See above 

Lines 190-200: SREP sounds amazing. I think we need more details about how the program fit within their high school coursework (during the week, did they go to the lab during school day?) In the summer, were they paid or just volunteers?

 The authors agree that information was lacking and have expanded on the “Program description” subsection under the Methods to include further details including relation to coursework, logistics of time, and compensation (Lines 222-268). 

More importantly (in terms of a selection bias issue): how were students selected for it? A selection system does introduce additional selection bias. The requirements of the program have been detailed as well as alternative options (Lines 242-254).

What are the sociodemographics of the student body at this school and in SREP? (line 274 is too late for this info to appear. The info is also vague there—report actual percentages).

 The authors have added available percentage data under Methods and the “Sources” subsection (Line 302-330, Table 2) to include the details on the race/ethnicity of the HB student body and those who participated in SREP. As noted (Line 298-300) HB does not collect information on household income, guardian educational background, or family structure so these demographics were unavailable.

I think the authors should pose research questions at the end of the introduction. Without doing that, it is hard to see how the different data sources and variables are going to be used (on pages 9-11). For example, i don’t know how supplemental survey which 25% of the cases responded too will be used.

 A “Purpose of the study” subsection has been added to the Introduction to pose research questions and goals of the study (Line 187-214).

In the data section, the authors also do not introduce how they will determine the control group or control population. Line 263 mentions some sources, but not specifically what they are and how they are used.

 The “Sources” subsection of the Methods has been expanded and now includes information on HB Non-SREP students as controls and lists the publicly accessible databases and tables that were used for comparison groups for each analysis (Line 355, Table 3). 

Line 268: the authors report that their alumni are not nationally representative, but they need to actually show the reader how they diverge from national stats so that we can use that information to evaluate the comparisons. So compare actual percentages between groups.

 The authors have acquired all demographic data available for HB SREP and Non-SREP students and have added available percentage data under Methods and the “Sources” subsection (Line 302-330) to include the details on the race/ethnicity. As noted in (Line 298-300) HB does not collect information on household income, guardian educational background, or family structure so these demographics were unavailable.

Line 271: says that the purpose is to assess SREP as a model. It seems very late in the paper to be stating that as the goal. Also, is that what the analyses really achieve? In lines 278-9: another goal is stated. The paper should have one goal (or a set of related goals) and it should be introduced early in the paper and then the analyses should support that goal).

 Authors agree that the goals stated throughout the paper appeared disparate and unrelated and need fine-tuned. Lines 403-408 (previously 278-9) have been removed and reworded in a “Purpose of the study” subsection in the Introduction that ties the educational and occupational outcomes assessed to the skill sets learned through the SREP to link bridging the confidence gap to a reduction in leakage in the STEM pipeline (Line 187-214). 

Section at line 282: this does not seem to be results, it seems like background on the program, i would move it back to the section on srep, unless you create new research questions that include this as one of the questions.

 Discipline dispersal in the SREP has been converted from Figure 1 to Table 1 (Line 270-279) and moved to the Methods under the “Program description” subsection to help characterize the SREP.

Line 305: this heading seems to be one of the key research questions. Pose it earlier.!

 Key research questions have been streamlined and related to achieve one overall goal. The research question is now posed earlier within the “Purpose of the study” subsection in the Introduction (Line 187-214). 

Line 316: is there a way to get these national stats for white and asian women alone? Since we know that minority students are less likely to go into stem careers, the national averages are lower than they would be for the group of students from this school. There might be better comparison stats than national averages when your students are not nationally representative.

 The authors appreciate the reviewer’s ideas and expertise on how to address the lack of ideal controls in this study. The authors have reviewed the publicly available databases and have added White female and Asian female subgroups as a better comparison than the national averages as suggested in all analyses possible (Lines 326-300). National averages were retained for wholistic comparisons.

Line 361: what does being “more firmly committed” mean? I would recommend just using language about switching majors in the headings for clarity.

 The language here is was unnecessarily confusing and the authors appreciate your recommendation for clarification. The wording has been changed to include switching majors (Line 479-482).

Line 377-378: i am not convinced that that gap could not be explained by the ses and race/ethnicity differences between the students in your sample and the national averages.

 The authors agree that this is a limitation of the study and have addressed the race/ethnicity differences by including demographic data on HB students (Table 2) and comparing to subgroups of White women and Asian women as a closer comparison. Unfortunately, HB does not keep records on SES (Line 297-300) and therefore this variable cannot be specifically controlled.

I would recommend posing a series of research questions (at the end of the intro) that lead the reader through the results.

 Key research questions have been streamlined and related to achieve one overall goal. The research questions are now posed earlier within the “Purpose of the study” subsection at the end of the Introduction (Line 187-214).

Line 568: the issues of selection bias in this paper are huge and since those who do srep are not representative in any way, i am just convinced that the national stats are the right comparison group. I think they could be utilized in the paper, but i don’t think they should be the featured story since the hb alumnae are not even close to representative (this was not reported directly, but i assume this, based on what the authors reported in the paper).

 The authors recognize the selection bias and have tried to be transparent about the issues in the Limitations section (Lines 733-748). We have also now included comparisons to HB Non-SREP students in all cases where there was data available (Table 2, Table 4, Fig 3) to do so.

One better comparison group would be hb students who did not do srep. I don’t see why (line 582) this group could not have been made available. That does not make sense to me, given that the study is already using alumni records. (if indeed that is the case, this should be better explained.)

 We have included comparisons to HB Non-SREP students in all cases where there was data available including demographics (Table 2), college majors awarded (Table 4)., advanced degrees (Fig 3A), and occupations (Fig 3B) and have explained why data is not available for other comparisons (Line 297-300).

The paper has a really cool set of data and i like it in many ways, but it is fundamentally flawed. Research looking at this question about the impact of program participation on stem trajectories would use propensity score matching to match each srep student with controls, conduct multivariate stats, and then that would isolate the effect of srep on their trajectory and you could really say that it helped and by how much it helped.

In the absence of that, it seems like making multiple comparisons is probably best. Comparing local stats for the community/city in which hb is located, comparing by race/ethnicity nationally, comparing to national averages, etc.

 The authors fully agree and would have preferred to use propensity score matching; however, as the data necessary for this type of statistical analysis is not available and cannot be collected retrospectively, the use of proportion test was the best fit (Line 392-394).

We appreciate your recommendation on alternative options given the circumstance. While we have kept the comparisons to national averages, we have used HB demographics to identify that our SREP group has a higher representation of Whites and Asians (Line 302-330, Table 2) and are therefore now including comparisons to national averages for White women and Asian women for as many analyses as possible (Table 2, Fig 1, Table 4, Fig 3). Community/city averages were not available. 

Comparing to national statistics only is not very meaningful, given that it appears this group is quite a bit more affluent and advantaged than the average american woman. One possibility is to look at all the stats by race, so find national stats for white woman and asian women and compare to them. However, since those stats for the hb alumni were not reported, i cannot provide definitive advice here on how to move forward. The current approach is problematic.

 All HB demographics that were available were pulled to identify representation in HB and within the HB SREP (Table 2). As there is a higher representation of Whites and Asians compared to the national population, we have included comparisons to national averages for White women and Asian women for as many analyses as possible (Line 328-330). 

The tables and figures are missing captions and notes. The reader needs to know the n of the groups and the source of the data in all of them. I also think basic tables are missing, such as descriptive stats on the hb alumni group (and how they compare to national stats, if those stats are used). Apologies for the missing captions and notes. Notes for tables and captions for figures were included within the document and figures were separate as instructed for submission. However, additional notes and sample size values have been added where they were missing previously. Table 2 was also added which includes the race/ethnicity demographics of the HB SREP and Non-SREP populations compared to the national population. Table 3 was added to summarize where all national stats originate and where they are used.

---

## [Editor Report · Decision Letter 1]

5 Oct 2021

A multi-year science research or engineering experience in high school gives women confidence to continue in the STEM pipeline or seek advancement in other fields: A 20-year longitudinal study

PONE-D-20-34867R1

Dear Dr. Miller,

We’re pleased to inform you that your manuscript has been judged scientifically suitable for publication and will be formally accepted for publication once it meets all outstanding technical requirements.

Kind regards,

Andrew R. Dalby, PhD

Academic Editor

PLOS ONE
---

## [Editor Report · Acceptance letter]

8 Oct 2021

PONE-D-20-34867R1 

A multi-year science research or engineering experience in high school gives women confidence to continue in the STEM pipeline or seek advancement in other fields: A 20-year longitudinal study 

Dear Dr. Miller:

I'm pleased to inform you that your manuscript has been deemed suitable for publication in PLOS ONE. Congratulations! Your manuscript is now with our production department. 

Kind regards, 

on behalf of

Dr. Andrew R. Dalby 

Academic Editor

PLOS ONE